# Researching the Influence of Rural University Campuses on Rural Economic Development: Evidence from Chinese Counties between 2001 and 2020

Cixian Lv *,†, Xiaotong Zhi †, Yuelong Ming, Kejun Zhang, Jia Sun, Haoran Cui and Xinghua Wang *

Normal College, Qingdao University, Qingdao 266071, China; zhixiaotong@qdu.edu.cn (X.Z.); mingyuelong@qdu.edu.cn (Y.M.); 15867856356@163.com (K.Z.); sunjia@qdu.edu.cn (J.S.); cuihaoran0527@163.com (H.C.)
* Correspondence: lvcixian@qdu.edu.cn (C.L.); xinghua_wang689@qdu.edu.cn (X.W.)
† These authors contributed equally to this work and share first authorship.

**Abstract:** While there have been studies on the relationship between higher education institutions and regional economic growth, few have delved into the economic impact of decentralized higher education institutions at the county level and associated reginal disparities in terms of socio-economic development. Utilizing the data of the Chinese universities that started to establish their campuses in counties since the year 1999, this study investigates the influence of rural university campuses on county-level GDP and industrial composition spanning from 2001 to 2020. It also delves into the temporal dynamics and regional discrepancies associated with this impact. The findings of this study show that (a) rural university campuses wield a notable positive influence on the GDP of their respective counties, particularly shaping the structure and ratio of secondary and tertiary industries; (b) the magnitude of this effect is contingent upon the duration of campus establishment and growth, intensifying over time; (c) variations in this impact are evident across the eastern, central, and western regions of China, where there are vast socio-economic differences. This study underscores the significant spillover effect of higher education decentralization on county-level economies and advocates for the pivotal role of rural university campuses in propelling county-level economic progress. Additionally, it proposes coordinated policy support from national, regional, and rural university campus authorities; the establishment of requisite support structures; and the comprehensive consideration of regional nuances.

**Keywords:** rural university campuses; decentralization of higher education; rural economic development; knowledge spillover effect

## 1. Introduction

The discourse surrounding the correlation between higher education and regional economic growth has undergone profound development in modern economics. Beginning with Schultz et al.'s human capital theory, and extending to the endogenous economic growth theories of Romer and Lucas, scholars have progressively illuminated the crucial roles of skills, knowledge, and innovation in fostering regional economic development [1–3]. Higher education stands out as not only the primary incubator for advanced knowledge and skilled talents but also a significant catalyst for regional innovation and technology diffusion, thereby exerting a substantial influence on economic growth. For instance, Andersson et al. demonstrated, using Sweden as a case study, how the Swedish government's investment strategy of decentralizing higher education across provinces has profoundly impacted the economies of regions hosting higher education institutions [4]. Similarly, Valero and van Reenen argue that regions with a higher density of universities per capita experience higher long-term per capita GDP growth rates, primarily due to the direct augmentation of a highly skilled labor force [5]. Many scholars posit that higher education's

potential contributions to regional development encompass enhancing regional enterprise productivity [6], nurturing regional human capital, upgrading regional infrastructure, stimulating local demand for goods and services, and enhancing employment opportunities and income levels for regional residents [7,8]. Consequently, higher education transforms local regions into hubs of high-quality talent and scientific knowledge, thereby generating significant knowledge spillover effects. Despite these insights, (a) existing research predominantly focuses on macroscopic levels, typically analyzing at national, provincial, or municipal scales, with limited attention paid to peripheral regions at the county level. Moreover, (b) studies rarely treat the establishment of university campuses in counties as natural experiments, thereby missing the opportunity to accurately analyze the dynamic, incremental impacts of county campuses on county-level economies. Furthermore, (c) existing research often fails to construct control groups for experimental interventions, leading to challenges in mitigating endogeneity issues in data models and achieving precise causal identification.

Since 1999, the Chinese government has implemented a policy of significantly increasing the enrollment of higher education institutions, known as the expansion policy of higher education enrollment. After the expansion, most university campuses in major cities have struggled to meet the demands brought about by the growth in student numbers, prompting them to seek further development space in the vicinity of cities. Meanwhile, counties with relatively developed economies around major cities are actively seeking new drivers such as knowledge, technology, and innovation to further develop their economies. Therefore, many Chinese higher education institutions have chosen to establish county campuses in economically developed areas surrounding major cities, starting from 1999. However, the following questions arise: Do the establishment and development of these rural university campuses significantly bolster economic growth in their host counties? Does this impact increase over time? Are there regional disparities in this impact across eastern, central, and western China? These questions warrant further investigation. Therefore, this paper aims to leverage data spanning from 2001 to 2020 to delve into the relationship between rural university campuses and county-level economic growth. The subsequent structure of this paper unfolds as follows: Section 2 encompasses a literature review and research hypotheses, Section 3 delineates the model settings and variable explanations, Section 4 presents empirical analysis and hypothesis testing, and Section 5 concludes with discussions.

## 2. Literature Review and Hypothesis Development

### 2.1. Overview of the Economic Sectors in Rural China

In rural China, economic activities typically revolve around agriculture, manufacturing, services, and emerging industries. Agriculture remains a fundamental sector, focusing on crop cultivation, animal husbandry, and agricultural enterprises. After 20 years of economic reforms, significant growth has been achieved in rural China. Despite concerns in the early 1990s about the slowdown in total factor productivity growth after a rapid increase in the early 1980s, by the late 1980s and 1990s, per capita food supply in China had reached levels similar to those of developed countries, making China one of the world's fastest-growing food exporting countries [9]. Economic activities in rural China are not limited to the productivity of the agricultural sector alone; in addition to traditional agriculture, the rural areas host an increasingly diverse range of agricultural processing industries and emerging commodity markets, contributing to value addition and rural employment [10]. Manufacturing activities in rural China encompass a range of industries including light industry, textiles, food processing, and small-scale production units. Despite attempts by the government to control the market over the past decade, agricultural markets have not only emerged but have also thrived, resembling markets in other countries more and more [11].

In addition, the transition of the population from rural to urban areas and from agriculture to industry is also central to the country's economic development [12]. Some scholars

consider the transformation of population structure in China in the 1980s lagged behind income levels [13]. However, in recent years, the transformation of the non-agricultural sector has been faster than any other sector. In the mid-1990s, barriers between rural and urban areas broke down, initiating an unprecedented and perhaps irreversible flow of labor to cities. Despite favorable macroeconomic conditions in the late 1990s, the surge in non-agricultural employment not only continued after 1995, but accelerated. Today, over 200 million people work outside farms, with over half of them working in cities. Nearly 85% of rural households have at least one member working in a non-agricultural sector [14].

Moreover, emerging industries such as renewable energy, e-commerce, and technology-based enterprises are gaining increasing attention, providing new opportunities for rural economic development [15]. In recent years, the rapid development of e-commerce in China's agricultural products has largely played a role in linking and matching the digital economy, accelerating the integration of agriculture and industrial services [16]. Digital finance in rural China not only enhances the production function of agriculture but also adds a leisure function, demonstrating good performance in the integration of agriculture and services [17]. However, at present, within the Chinese government, there is a lack of colleagues who understand information technology, possess management skills, and have rich rural work experience, and there is a shortage of suppliers in China's digital agricultural construction. The development level of information infrastructure in different regions is unbalanced and the construction of data-sharing systems lags behind, severely hindering the integrated development of rural industries and limiting the driving force of the digital economy for urban–rural integration [18,19].

### 2.2. Decentralization of Higher Education

Since the 1960s, higher education enrollment across most Western nations has undergone substantial and sustained growth. Simultaneously, there has been a trend towards regional diversification in higher education, aiming to distribute universities to peripheral towns and regions surrounding cities [20]. Rossi and Goglio found that since the early 21st century, British universities have established satellite campuses, whereby established universities located in major cities have dispersed some university functions to previously university-free peripheral urban areas [8]. Over this period, many British universities have also established secondary campuses, with some in metropolitan areas and others in small towns and rural locales [20]. The establishment of satellite campuses by Australian universities likewise stems from the ongoing surge in demand for higher education nationwide and concerns regarding educational equity. Research indicates a significant positive correlation between the decentralization of higher education and regional economic development within the country [21]. Since the 1990s, Italian universities have developed satellite campuses on the outskirts of cities to cater to the needs of surrounding communities and stakeholders, who often shoulder most of the construction costs [22]. Norway's decentralization of higher education dates back to the mid-1950s when the government implemented policies to enhance the geographical spread of higher education within the country [23]. Similarly, the Constitution of Canada allocates educational responsibility to the provinces, primarily by expanding the existing universities, granting autonomy to establish regional campuses in urban peripheries, and establishing institutions in areas with insufficient higher education services to expand their geographical coverage [24]. Similar scenarios are also evident in the United States, where universities often establish rural community colleges in rural areas as smaller, rural, two-year branches of large state universities [25]. In China, following a significant increase in enrollment in 1999, universities have gradually commenced operations in county-level areas around cities. Existing universities establish campuses in new regions, typically (though not always) in peripheral urban areas previously lacking higher education, representing a new form of decentralization in global higher education.

*2.3. Research Hypotheses on the Relationship between Decentralization of Higher Education and Regional Economic Development*

There exists a causal relationship between the decentralization of higher education and regional economic development. Cantoni and Yuchtman examined the impact of the birth of the first universities after the Great Schism in 1386 on the commercial revolution in medieval Germany, finding that regions closer to universities experienced faster establishment rates of commercial markets, validating the causal role of medieval universities in expanding economic activities in their locations [26]. Boucher et al. argue that campuses in peripheral areas are more important for the development of the region than universities in metropolitan cores, being more likely to engage in economic and social activities in the region [27]. Bleaney et al. suggest that dispersed campuses located in counties or rural areas can attract students, scholars, technical managers, and various types of visitors, which can stimulate local industries such as goods and services [28]. Benneworth and Charles propose that decentralized higher education institutions can enhance local business productivity by providing qualified labor, scientific knowledge, advanced scientific equipment, and consulting services [29]. These decentralized institutions build up territorial knowledge pools, create related technology spin-off companies to increase local entrepreneurial activities, and create high-tech, high-paying job opportunities. Additionally, decentralized institutions in counties or rural areas can serve as bridges between local communities and the external world, becoming key centers in local development policies in peripheral areas [30]. Kantor and Whalley, using instrumental variable strategies to address the endogeneity of universities in their regions, still found evidence of a causal relationship between universities and higher productivity in their counties [31]. Moreover, these centers can generate significant synergy effects and benefits for the local region, potentially accessing more national funding for scientific, educational, and innovative activities in that region. Hence, this study proposes the following research hypothesis:

**Hypothesis 1 (H1).** *Rural university campuses significantly drive economic growth in their respective counties (including GDP, the value added of primary, secondary, and tertiary industries, completed fixed investment, and general public budget revenue, etc.); they promote adjustments in county industrial structures, particularly fostering the development of secondary and tertiary industries.*

The causal relationship between the decentralization of higher education and regional economic development is subject to temporal effects. Charles studied the relationship between the establishment of rural campuses by six universities in the UK and local innovation activities, finding that although these campuses could integrate into regional innovation systems, they faced many difficulties in terms of economies of scale and scope [20]. There is a need for long-term strategic planning to achieve integration with local economic development. Cermeño studied the impact of county-level universities in the United States from 1931 to 1980 on population density, GDP, and market size in the local counties [32]. The study suggested a generally positive and significant impact of universities on county population density and GDP growth, with counties hosting universities experiencing annual growth rates between 1% and 3%. However, without additional investment, these spillover effects eventually disappear. The construction and development of rural university campuses themselves, as well as the coupling of skills, knowledge, and innovation with local industries, also require time. As time progresses, the knowledge spillover effects of rural university campuses will be further validated. Thus, this study proposes the following research hypothesis:

**Hypothesis 2 (H2).** *The impact of rural university campuses on county economic development will gradually strengthen over time.*

The causal relationship between the decentralization of higher education and regional economic development exhibits certain heterogeneity. Recent studies have questioned

the contribution of campuses to the economic development of their regions, suggesting a disconnect between these campuses and local entrepreneurship and innovation ecosystems [33]. Due to their regional constraints, they are unable to benefit from knowledge spillovers from academic research [34]. Some empirical evidence indicates that graduates from county or rural campuses perform better in terms of job contracts, salary levels, etc., compared to graduates from main campuses, possibly because these campuses offer curricula more closely aligned with local economic development and industry needs [35]. Considering the significant differences in initial industrial bases, human capital accumulation, and degree of marketization among different regions in China, local governments have significant differences in policies regarding rural university campuses. Hence, this study proposes the following research hypothesis:

**Hypothesis 3 (H3).** *The effects of rural university campuses on county economic development exhibit regional heterogeneity.*

### 3. Methodology

*3.1. Data Sources*

This study utilized panel data spanning 20 years from 2001 to 2020 from nine cities across China, including Qingdao, Hangzhou, and Guangzhou in the east, Wuhan, Changsha, and Zhengzhou in the central region, and Chongqing, Chengdu, and Xi'an in the west, as the analytical sample. Data sources were primarily derived from the Statistical Yearbooks of related provinces in China, while information related to rural university campuses was compiled from publicly available data provided by the educational bureaus of the provinces. During the data collection process, efforts were made to cross-validate statistical methodologies employed in different data sources. This study collected a total of 58 counties and 156 rural university campuses from the 9 cities, representing 53.8% of all rural university campuses nationwide and ensuring a representative sample. These rural university campuses generated 1908 data points in total spanning from 2001 to 2020.

*3.2. Research Model*

This study primarily employed the Stata software 17.0 for data analysis. We initially employed a time-varying Difference-in-Differences (DID) model due to variations in the timing of the establishment of university campuses in counties across different cities in China's central and western regions, as well as differences in the number of county-level campuses in each city. Traditional double-difference methods generally apply to situations where the sample status remains unchanged at the same time point. Additionally, we used event-study analysis to observe the differences in effects before and after the implementation of rural university campuses in two sample groups. This analysis also captured dynamic changes over time and validated Hypothesis 2. Finally, synthetic control methods were employed to determine weights through data-driven approaches, reducing subjective selection errors and avoiding endogeneity issues in policies.

Considering the varied construction timelines and rates of progress among rural university campuses throughout China, a time-varying DID model with broader criteria was chosen. In contrast to the standard DID model, which assumes the uniform initiation of policy shocks across all entities, this model accommodates the varied timelines of policy implementation. To test research H1, the initial model is established as follows:

$$\text{Growth}_{nt} = \delta + \alpha \text{Campus}_{nt} + \rho' X_{nt} + \theta_n + \lambda_t + \varepsilon_{nt} \tag{1}$$

In Model 1, subscripts *n* and *t* represent the counties hosting rural university campuses and the particular year when the campuses were built, respectively. The dependent variable, *Growth*$_{nt}$, refers to the comprehensive economic growth and the influence of various sectoral economic indicators within a specific county (denoted as "*n*") in a particular year (denoted as "*t*"). This includes factors such as the county's GDP, value added in primary, secondary, and tertiary industries, completed fixed asset investment, and general public

budget revenue, among others. This study investigates the structural differences in rural economic development influenced by rural university campuses, particularly the proportions of secondary and tertiary industries. To address heteroscedasticity, all variables are logarithmically transformed.

The primary explanatory variable, *Campus*, reflects rural university campuses and is measured in two forms: (1) Quantity of operating rural university campuses in county $n$ in year $t$—this refers to the number of rural university campuses that are actively operating within a specific county (denoted as $n$) during a particular year (denoted as $t$). These campuses are the ones that are open and functioning, providing educational services and engaging in various activities related to their mission. (2) Cumulative operational years of rural university campuses in county $n$—this represents the total duration for which rural university campuses have been operational within a specific county ($n$) up to the current year ($t$). It is the sum of the years since each campus was established and began its operations in the county. For example, if a campus was established five years ago and has been operating continuously since then, its cumulative operational years would be five. Consequently, the product of the number of rural university campuses and their operational years is introduced as a new explanatory variable to mitigate differences in operational duration and obtain a more accurate average treatment effect. Unlike standard DID models employing interaction terms, *Campus* varies over time and across entities, representing a treatment variable.

Control variables $X$ encompass factors varying over time and across entities, including population density (year-end registered population/area under jurisdiction) and the number of students enrolled in primary and secondary schools in each county. Furthermore, adjustments were made to address discrepancies in economic statistical data calibers across certain years, incorporating dummy variables to identify the years of statistical caliber changes. These adjustments aim to mitigate the influence of statistical caliber differences, which manifest in both inflation and deflation scenarios. County fixed effects $\theta_n$ and time fixed effects $\lambda_t$ are included to account for inherent socio-economic attributes of counties and year-specific fluctuations, respectively.

According to H2, this paper assumes that the impact of rural university campuses on the local county economy varies over time. To address this, this study references the methodology of Beck et al. and further integrates the initial time-varying DID model with event-study methods to capture the changing trends in the effects before and after the implementation of rural university campuses [25,36]. Drawing from the approach of Jacobson et al. [26,37], the event-study model equation is set as shown in Model (2):

$$\text{Growth}_{nt} = \delta + \sum_{m, m \neq o} \gamma_m D_{nt}^m + \beta' X_{nt} + \theta_n + \lambda_t + \varepsilon_{nt} \tag{2}$$

By introducing a set of binary variables, this paper estimates the impact of rural university campuses at each time point before and after implementation, thereby capturing the fundamental trends of these intervention events over time. If county sample $n$ adopted the campus in year $t - m$, the value is set to 1. Thus, each binary variable represents the estimated effect of counties in the experimental group adopting the campus in the $m$th year, essentially expanding upon the initial time-varying DID model. Lastly, to validate research H3, this paper introduces interaction terms to examine the differences in the impact of rural university campuses between eastern, central, and western regions of China.

### 3.3. Variable Description

The indicators and data sources for each variable in the models are summarized in Table 1.

**Table 1.** Indicators and data sources.

| Variables | Descriptions | Sources |
|---|---|---|
| Growth | Logarithm of GDP (in hundred million RMB) | "Yearbooks" and "Statistical Yearbooks" of relevant provinces in China |
| | Value added of the primary industry | |
| | Value added of the secondary industry | |
| | Value added of the tertiary industry | |
| | Completed fixed investment | |
| | General public budget revenue | |
| Campus | Number of rural university campuses in particular year | Publicly available data from relevant provincial educational bureaus of China |
| | Number of operating years of rural university campuses | |
| D | Whether rural university campuses were in use in that particular year | |
| X | Logarithm of the number of students enrolled in primary and secondary schools (in ten thousand people) | "Yearbooks" and "Statistical Yearbooks" of relevant provinces in China |
| | Population density (per hundred people per square kilometer) | |
| Other variables | Whether the county is located in the eastern region of China | Publicly available data from relevant provincial educational bureaus of China |
| | Whether the county previously had a campus | |

### 3.4. Endogeneity Issues and Robustness Checks

To address potential endogeneity issues arising from factors such as omitted variables, measurement errors, and bidirectional causality among variables in this study, we drew insights from Abadie et al. and utilized the synthetic control method to mitigate interference from other confounding factors [38]. This method was utilized to assess the impact of knowledge spillovers from land-grant universities in the United States on regional production efficiency and measurement techniques. Additionally, we referenced historical events such as the decentralization reforms in spatial layout undertaken by Swedish universities as natural experiments to minimize interference from endogenous factors [4]. By integrating causal inference evaluation methods including double differences, fixed effects, and synthetic control, we weighted the control group counties to address endogeneity issues between rural university campuses and rural economic development. As a non-parametric approach, the synthetic control method extended traditional DID models, thereby increasing the likelihood of fulfilling the common trend assumption of DID models. By determining weights through data-driven methods, subjective selection errors were minimized, and policy endogeneity issues were avoided.

## 4. Results

### 4.1. Descriptive Results

Table 2 presents the descriptive statistics of the variables involved in the models. Columns 2, 3, and 4 of Table 2, respectively, provide the descriptive results for the control group, treatment group, and the entire sample, with the treatment group comprising 723 samples, accounting for 37.9% of the total sample size. From the descriptive statistics of the dependent variables, significant differences are observed among the groups, with the mean values of each explanatory variable in the treatment group surpassing those of the control group.

**Table 2.** Descriptive results.

| Variables | M (SD) | | |
|---|---|---|---|
| | **Control Group** | **Treatment Group** | **All Samples** |
| Dependent variable GDP logarithm | 4.965 (1.173) | 5.381 (0.932) | 5.012 (1.121) |
| Value added of the primary industry | 1.823 (1.673) | 2.312 (0.823) | 1.975 (1.123) |
| Value added of the secondary industry | 3.983 (1.256) | 4.723 (0.875) | 4.103 (1.231) |
| Value added of the tertiary industry | 4.562 (1.321) | 5.321 (1.263) | 4.891 (1.256) |
| Completed fixed investment | 4.561 (1.162) | 5.238 (1.230) | 4.893 (1.256) |
| General public budget revenue | 2.451 (1.123) | 2.985 (1.087) | 2.762 (1.116) |
| Independent variable | | | |
| Number of rural university campuses in particular year | 0 (0) | 1.823 (2.321) | 0.382 (1.356) |
| Number of operating years of rural university campuses | 0 (0) | 14.26 (23.12) | 3.89 (11.93) |
| Whether rural university campuses were in use in that particular year | 0 (0) | 2 (2.125) | 0.126 (0.613) |
| Control variables | | | |
| Logarithm of the number of students enrolled in primary and secondary schools | 2.123 (0.532) | 2.356 (0.582) | 2.219 (0.351) |
| Population density (per hundred people per square kilometer) | 47.012 (79.12) | 23.563 (25.123) | 39.287 (73.351) |
| Whether the county is located in the eastern region of China | 0.325 (1.026) | 0.339 (1.281) | 0.387 (0.597) |
| Whether the county previously had a campus | 0.387 (0.871) | 0.621 (1.161) | 0.456 (0.738) |
| Number of observations | 1185 | 723 | 1908 |

*4.2. Baseline Regression Analysis*

Table 3 presents the results of the baseline regression analysis. The dependent variables in each column of the table include the logarithmic forms of GDP, the value added of the primary industry, the value added of the secondary industry, the value added of the tertiary industry, and general public budget revenue. From the regression results, it is evident that all model equations exhibit a good explanatory power. Specifically, rural university campuses have a significantly positive impact on the overall GDP growth of their respective counties (with a regression coefficient of 0.0352), and this effect is particularly pronounced in the growth of the secondary industry value added (with a regression coefficient of 0.0536). The effects of completed fixed investment and the value added of the tertiary industry are also notable (with regression coefficients of 0.0361 and 0.0317, respectively). This may be attributed to the fact that the establishment and development of rural university campuses have driven the industrialization level of their respective counties. Additionally, the various goods and services activities brought about by the large-scale student and staff populations, as well as their families, have also propelled the development of the tertiary industry. Therefore, research hypothesis H1 has been partially validated.

**Table 3.** Baseline regression results.

| | GDP | Value Added of the Primary Industry | Value Added of the Secondary Industry | Value Added of the Tertiary Industry | Completed Fixed Investment | General Public Budget Revenue |
|---|---|---|---|---|---|---|
| Independent variables Number of rural university campuses | 0.0352 ** (0.022) | 0.0189 (0.016) | 0.0536 *** (0.028) | 0.0317 * (0.026) | 0.0361 * (0.037) | 0.0071 (0.019) |
| Year fixed effect | yes | yes | yes | yes | yes | yes |
| County fixed effect | yes | yes | yes | yes | yes | yes |
| Number of observations | 1908 | 1908 | 1908 | 1908 | 1908 | 1908 |
| R-square | 0.376 | 0.153 | 0.218 | 0.362 | 0.253 | 0.137 |

Notes: Robust standard errors clustered by county are reported in parentheses. Statistical significance denoted by *** $p < 0.01$, ** $p < 0.05$, and * $p < 0.1$.

### 4.3. Time Effects Analysis

The coupling degree between county-level campuses and the economic and social development of their respective counties is an ongoing process. Therefore, this paper further examines the influence of rural university campuses on their respective counties in the fourth and sixth years after establishment, as shown in Table 4. A cross-sectional comparison reveals that the results in Table 4 are generally consistent with those in Table 3. However, upon longitudinal analysis, it is observed that, over time, there is a noticeable increase in the regression coefficients of equations with the logarithm of GDP, the value added of the secondary industry, and the value added of the tertiary industry as the dependent variables. This indicates that the knowledge spillover effects of rural university campuses gradually strengthen over time. Thus, research hypothesis H2 is validated.

**Table 4.** Comparison at different time points.

| | GDP | Value Added of the Primary Industry | Value Added of the Secondary Industry | Value Added of the Tertiary Industry | Completed Fixed Investment | General Public Budget Revenue |
|---|---|---|---|---|---|---|
| Independent variables Rural university campuses (Year 1) | 0.0352 ** (0.022) | 0.0189 (0.016) | 0.0536 *** (0.028) | 0.0317 * (0.026) | 0.0361 * (0.037) | 0.0071 (0.019) |
| Rural university campuses (Year 2) | 0.0451 * (0.018) | 0.0156 (0.017) | 0.0589 ** (0.011) | 0.0391 * (0.023) | 0.0325 (0.029) | 0.0129 (0.018) |
| Rural university campuses (Year 3) | 0.0560 ** (0.013) | 0.0167 (0.013) | 0.0621 * (0.023) | 0.0420 * (0.019) | 0.0328 * (0.021) | 0.0135 (0.021) |
| Year fixed effect | yes | yes | yes | yes | yes | yes |
| County fixed effect | yes | yes | yes | yes | yes | yes |
| Number of observations | 1908 | 1908 | 1908 | 1908 | 1908 | 1908 |
| R-square | 0.221 | 0.138 | 0.107 | 0.257 | 0.219 | 0.196 |

Notes: Due to space constraints, this table only reports the regression results for the independent variable of rural university campuses in the first, fourth, and sixth years after establishment, with the basic setup of each regression equation consistent with Table 3. Robust standard errors clustered by county are reported in parentheses. Statistical significance is denoted by *** $p < 0.01$, ** $p < 0.05$, and * $p < 0.1$.

### 4.4. Analysis of Regional Heterogeneity

In comparison to the central and western regions, the eastern region of China exhibits a higher level of economic development, with greater flexibility and diversity in the flow of talent, information, and capital. To examine the regional disparities influenced by rural university campuses, this study introduces an interaction term between the number of years of operation of rural university campuses and the eastern region as an explanatory variable (see Table 5). We introduced the interaction term "rural university campuses (Year 1/4/6) × eastern area" to examine the differential effects of rural university campuses between the eastern, central, and western regions of China. Furthermore, by incorporating a time axis, this study examined the long-term variations of these effects. The sign and magnitude of the interaction term could help us understand the interaction between these variables. If the interaction term of two independent variables is positive, then their combined effect will be greater than the sum of their individual effects. Regression results confirm this

hypothesis, with positive coefficients for all interaction terms, and statistically significant coefficients for the logarithms of GDP, the value added of the secondary industry, and the value added of the tertiary industry. In this context, the knowledge spillover effects of rural university campuses in counties in the eastern region are significantly higher than those in counties in the central and western regions. Thus, research hypothesis H3 is validated.

**Table 5.** Results of regional difference analysis.

| | GDP | Value Added of the Primary Industry | Value Added of the Secondary Industry | Value Added of the Tertiary Industry | Completed Fixed Investment | General Public Budget Revenue |
|---|---|---|---|---|---|---|
| Independent variables | | | | | | |
| Rural university campuses | 0.0051 ** | 0.0012 | 0.0069 ** | 0.0058 * | 0.0101 | 0.0029 |
| (Year 1) × eastern area | (0.002) | (0.004) | (0.002) | (0.003) | (0.002) | (0.003) |
| Rural university campuses | 0.0062 * | 0.0009 | 0.0076 * | 0.0059 | 0.0123 | 0.0023 |
| (Year 4) × eastern area | (0.003) | (0.003) | (0.001) | (0.004) | (0.003) | (0.003) |
| Rural university campuses | 0.0070 ** | 0.0016 | 0.0081 | 0.0074 * | 0.0135 | 0.0027 |
| (Year 6) × eastern area | (0.002) | (0.005) | (0.002) | (0.002) | (0.001) | (0.002) |
| Year fixed effect | yes | yes | yes | yes | yes | yes |
| County fixed effect | yes | yes | yes | yes | yes | yes |
| Number of observations | 1908 | 1908 | 1908 | 1908 | 1908 | 1908 |
| R-square | 0.220 | 0.111 | 0.234 | 0.251 | 0.205 | 0.112 |

Notes: Due to space constraints, this table only reports the regression results for the independent variable of rural university campuses in the first, fourth, and sixth years after establishment, with the basic setup of each regression equation consistent with Table 3. Robust standard errors clustered by county are reported in parentheses. Statistical significance is denoted by ** $p < 0.05$, and * $p < 0.1$.

## 5. Discussion

This study utilized a time-varying DID model to analyze the causal impact of the establishment and development of rural university campuses on various economic indicators within their respective counties in China. Additionally, an event-study model was employed to examine the historical average trends in rural economic development associated with rural university campuses. Furthermore, the interaction analysis was utilized to further verify the regional heterogeneity of this impact across the eastern, central, and western regions.

Overall, the willingness of graduates from urban universities to return to county areas for employment or entrepreneurship was not high. The empirical data of this study showed that county-level universities could directly and indirectly increase high-quality labor force for county industries. This research conclusion corresponded with the human capital theory proposed by Schultz et al., (1962), as well as the endogenous economic growth theories of Romer (1986) and Lucas (1988), confirming the role of human capital in regional economic growth. At the same time, the empirical data of this study validated the existence of a certain causal relationship between the decentralization of higher education and regional economic development [1–3]. This viewpoint was consistent with the research conclusions of Bleaney et al., (1992), Benneworth and Charles (2005), and Cantoni and Yuchtman (2014), among others [26,28,29]. However, this study further expanded on the significant temporal effects and regional heterogeneity in the causal relationship between the decentralization of higher education and regional economic development.

Firstly, our findings reveal that the establishment and development of rural university campuses exerted a notable positive influence on the overall economic growth of their respective counties. This impact was primarily reflected in driving growth in the secondary and tertiary sectors, particularly affecting the demand for knowledge-intensive business services. The study identified a significant causal relationship between rural university campuses and county-level economies, corroborating previous research [8,39]. Moreover, it aligns with prior studies suggesting that satellite campuses of universities made specific contributions to the economic development of their regions, with such impact effects unlikely to materialize without them. While the academic and technological innovation capacities of many county-based university campuses may be relatively modest, they often contribute to county development through avenues such as cultural activities, fostering social inclusivity, and commercial activities [40,41]. Notably, these rural university campuses

also provide various talent training programs essential for regional enterprise development, such as vocational education. Furthermore, the cultural, sports, and recreational activities offered by these campuses contribute to enhancing the attractiveness of local areas. However, this study suggests that the impact of rural university campuses on the agricultural sector was not significant, consistent with the findings of Kline and Moretti, which indicate that the knowledge spillover effects of universities on the agricultural sector are relatively small [31,42].

Secondly, our research indicates that with the increase in years of operation, rural university campuses exhibited a gradually strengthening effect on the overall economy of their respective counties. This finding is consistent with the results of Charles and Cermeño, suggesting a significant time effect in the causal relationship between rural university campuses and county-level economies [20,32]. Fundamentally, the significant role of rural university campuses at the regional level continues to stem from their creation of human capital and the high skills and employability of their graduates [43,44]. The full realization of the effects of these newly trained labor forces entering the market requires a considerable amount of time, consistent with Moretti's multiplier effect, which posits that the impact of new ideas and technologies generated by universities on knowledge-intensive services follows a sustained time frame [45].

Thirdly, this study reveals that compared to rural university campuses in the central and western regions of China, those in the eastern region had a more pronounced effect on the overall economic growth of their respective counties. Previous research has found that regions with weaker economic foundations in the UK often lack institutions engaged in basic research, while the most economically developed southern regions are home to many universities and their academic research institutions. This aligns with recent qualitative evidence regarding the weak demand for regional higher education knowledge-based services and activities in the Welsh business community [46]. The endowment of natural elements in a region often determines the level of human capital, with knowledge spillover effects in former mining communities relatively low [47,48]. Therefore, rural university campuses achieved multiplier effects through graduate employment in counties and support for technological innovation in local enterprises, within the constraints of regional industrial foundations and natural resources in the eastern and western regions.

One contribution of this study is the theoretical proposition that rural university campuses are primary drivers of county-level economic development. While the overall level of local attachment and entrepreneurial willingness among university graduates remains relatively low [49], county-level universities play a significant role in augmenting the local labor force directly and indirectly. Studies indicate that the prosperity of talent serves as a critical driver for the comprehensive development of rural industries [50]. Human capital emerges as a potent force in facilitating the integration of rural industries [51]. Mechanism studies reveal that in areas with low levels of rural human capital and correspondingly lower cultural proficiency among rural laborers, there exists a challenge of limited internet awareness and digital technology application capabilities. This impedes rural laborers in accessing vital information on agricultural industry development through digital platforms and complicates the seamless integration of agricultural digital technology with other sectors. Conversely, regions with higher levels of rural human capital and enhanced cultural proficiency among rural laborers are better equipped to leverage agricultural digital technology and gather agricultural resource information, thus fostering the holistic development of rural industries [52]. Previous literature has predominantly explored the economic effects of higher education from a macro perspective. From the standpoint of regional policy, a wealth of research indicates that rural university campuses can promote regional scientific, technological, and innovative capabilities, thereby driving regional economic development [53,54]. Leveraging the natural experiment of China's higher education institutions establishing campuses in counties since 1999, this study examines the knowledge spillover effects of higher education decentralization on county-level economies from

dynamic and micro perspectives and proposes the theory that rural university campuses are drivers of county-level economic development.

Another contribution of this study is the design and effective evaluation of the measurement tools for the impact of higher education institutions on county-level economic development, overcoming endogeneity issues in the data model. This study dedicated substantial efforts to gathering 1908 samples from 156 rural university campuses, spanning the period from 2001 to 2020. It implemented a dynamic time-varying DID model, utilized a comprehensive control methodology, and deployed a combination of synthetic control methods and event-study analysis, alongside relevant variable configurations. These approaches successfully addressed endogeneity concerns within the data model, facilitating a thorough assessment of the influence of higher education institutions on county-level economic advancement. This research methodology is validated by Liu's study [55], which used the U.S. land-grant universities as examples, avoiding endogeneity by constructing comprehensive control groups similar to treatment groups and proving that these schools not only increased population density in recipient counties but also allowed knowledge effects to spill over into other economic sectors.

Finally, the findings of this study resonate strongly with the notion of rural resilience, as they underscore the multifaceted contributions of rural university campuses to county-level economies. By fostering human capital development, promoting entrepreneurship, and facilitating knowledge transfer, these campuses bolster the resilience of rural communities in the face of economic challenges. Moreover, the observed temporal effects and regional heterogeneity in the relationship between the decentralization of higher education and regional economic development shed light on the nuanced dynamics shaping rural resilience across different geographical contexts.

## 6. Policy Implications

In order to further enhance the positive impact of rural university campuses on county-level economic development, this study proposes the following policy recommendations. Firstly, it is recommended that national and regional governments and rural university campuses jointly introduce coherent supportive policies. Additionally, policies should be dynamically adjusted based on the knowledge spillover effects of county-level campuses and the status of regional economic development to ensure the sustainability of their effects. Policies should consider how to encourage collaborative research formats to establish a common understanding among academia, rural small and medium-sized enterprises, civil society, and government. Specialized systems related to the distinctive development of rural university campuses should be promptly introduced to clarify the development goals and positioning of rural university campuses, as well as delineate relevant institutional frameworks for their distinctive development, thereby providing theoretical frameworks and practical bases for promoting characteristic development in rural areas and rural university campuses. As noted by Laranja et al. [48], if policies at the national level are too fragmented and lack coherence, regional universities may struggle to become centers of science, technology, and innovation in their respective areas. For university administrators and policymakers, it is essential to recognize that besides personalized benefits for each student, community interests constitute a significant aspect of these campuses. While every student benefits from their higher education experience, the model of rural university campuses attempts to determine how communities can also benefit alongside students, even if students leave the area upon graduation. Student knowledge exchange becomes an additional avenue to create broader community value, fostering learning among community members, building social capital and community resilience, and ensuring sustained benefits.

Secondly, national and regional governments are advised to establish adequate supporting institutions around rural university campuses. These institutions include research organizations providing public services, key laboratories dedicated to applied research, and a large number of enterprises capable of commercializing and providing services. We

should increase investment in rural university campuses; actively promote the development advantages of rural university campuses; integrate rural university campuses into the public service system of large, medium, and small cities, appropriately expanding the overall management authority of rural university campuses; increase funding for rural university campuses; and implement strong–weak paired assistance policies. To address discrepancies in educational concepts and operating methods, developed provinces can assist rural university campuses in other provinces, encouraging provinces (regions, municipalities) to organize local universities to conduct custodial assistance work to achieve win–win cooperation. Evidence from leading regions in knowledge spillover effects (such as Silicon Valley in the U.S.) suggests that universities need comprehensive supporting institutions to maximize their impact [31]. Rural university campuses are akin to subsidiaries of multinational corporations. These subsidiaries need to be effectively integrated into regional economic activities and become integral parts of the entire regional innovation supply chain.

Thirdly, it is suggested that national and regional governments introduce appropriate rural university campuses based on regional disparities in local economic development, industrial structure, population density, and other factors. Previous research indicates that rural university campuses are emerging as a new form of global higher education decentralization and effectively driving county-level economic development. However, it is important to note that the knowledge spillover effects of these campuses vary significantly across regions and do not occur uniformly across all jurisdictions. The government needs to consider the overall situation, refine its top-level design, and formulate sound policies to guide the integrated development of rural university campuses and local economic industries based on regional disparities and particularities. It should establish a community of rural university campuses, set up demonstration areas and schools, actively encourage mutual learning and communication among regions, rationally plan the coordinated construction of rural university campuses between regions, encourage the sharing of similar regional information and resources, and facilitate win–win cooperation between complementary regions, thus ensuring the smooth flow of knowledge capital and technology among regions.

In sum, the proposed policies stemming from this study's findings are poised for prompt implementation, with initial planning and coordination among national and regional governments and rural university campuses expected to span 6–12 months. Success could be gauged through a diverse array of metrics, including economic growth indicators, graduate return rates, collaborative efforts, and enhancements in talent retention and community engagement. Stakeholders, ranging from governments to rural universities, academia, small and medium-sized enterprises, and civil society, each have distinct yet interrelated roles in policy formulation, implementation, and collaboration, necessitating the clear delineation of responsibilities. Drawing upon successful case studies from other regions, such as community engagement initiatives for rural revitalization by rural universities in Scotland, rural development programs in Scandinavia, and initiatives in rural regeneration in the United States, could provide valuable insights and inspirations for effective strategies, challenges encountered, and lessons learned. However, addressing coordination hurdles between stakeholders, ensuring equitable resource distribution, and adapting policies to diverse regional contexts are key challenges that require mitigation strategies like regular evaluations, capacity building, and targeted interventions. Additionally, fostering awareness, stakeholder engagement, and flexibility in policy design and implementation will be vital in navigating potential obstacles and ensuring the success and sustainability of the proposed policies.

## 7. Limitations and Future Research

There are two main limitations to this study. Firstly, the sample selected only covered rural university campuses in nine major cities in the eastern, central, and western regions of China, which may lack full representativeness. It is hoped that future research can

explore more extensively with higher-quality data. Secondly, existing studies have shown that the effects of different types of county-level campuses, such as intra-city relocation, construction in different locations, and synchronous education between urban and rural areas, may also vary. This study did not fully account for these potential confounding factors and will be improved upon in subsequent research.

**Author Contributions:** Conceptualization, C.L. and X.W.; Data curation, X.Z., K.Z. and H.C.; Formal analysis, Y.M., K.Z. and H.C.; Funding acquisition, C.L.; Investigation, X.Z.; Methodology, X.Z. and Y.M.; Project administration, C.L.; Resources, J.S.; Software, C.L. and X.Z.; Supervision, C.L. and X.W.; Validation, J.S.; Visualization, X.Z.; Writing—original draft, Y.M.; Writing—review and editing, C.L., K.Z., J.S. and X.W. All authors have read and agreed to the published version of the manuscript.

**Funding:** This research was funded by National Social Science Foundation (Education) Project, grant number BIA190164.

**Institutional Review Board Statement:** This project was approved by the Ethics Committee of Qingdao University.

**Informed Consent Statement:** Not applicable.

**Data Availability Statement:** This study employed publicly available data sourced from the Statistical Yearbooks of nine provinces (see Section 3.1), along with data from their respective educational bureaus.

**Conflicts of Interest:** The authors declare no conflicts of interest.

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
