# Peer review of "Researching the Influence of Rural University Campuses on Rural Economic Development: Evidence from Chinese Counties between 2001 and 2020"

_sustainability, doi:10.3390/su16103974_

Round 1

Reviewer 1 Report

Comments and Suggestions for Authors

Overall this manuscript is acceptable. It is best to improve the discussion section by increasing the scientific sharpness of the results found.

In the results section (lines 320-393), expand your discussion and relate your findings more strongly to a) theory, b) empirical, c) context; and d) highlight its economic, academic/research, and policy implications. Closely relate and cite the papers you have presented in the background, theory, and empirical literature review sections to the findings you present here.

Author Response

Reviewer 1

Comments and Suggestions for Authors

Overall this manuscript is acceptable. It is best to improve the discussion section by increasing the scientific sharpness of the results found.

In the results section (lines 320-393), expand your discussion and relate your findings more strongly to a) theory, b) empirical, c) context; and d) highlight its economic, academic/research, and policy implications. Closely relate and cite the papers you have presented in the background, theory, and empirical literature review sections to the findings you present here.

Response:

We appreciate the suggestions provided by the reviewers on this point. Accordingly, we have made the following revisions:

Overall, the willingness of graduates from urban universities to return to county areas for employment or entrepreneurship was not high. The empirical data of this study showed that county-level universities could directly and indirectly increase high-quality labor force for county industries. This research conclusion responded to the human capital theory proposed by Schultz et al. (1962), as well as the endogenous economic growth theories of Romer (1986) and Lucas (1988), confirming the role of human capital in regional economic growth. At the same time, the empirical data of this study validated the existence of a certain causal relationship between the decentralization of higher education and regional economic development. This viewpoint was consistent with the research conclusions of Bleaney et al. (1992), Benneworth and Charles (2005), Cantoni and Yuchtman (2014), among others. However, this study further expanded on the significant temporal effects and regional heterogeneity in the causal relationship between the decentralization of higher education and regional economic development.” (see page 11)

Reviewer 2 Report

Comments and Suggestions for Authors

The relationship between higher education and regional economic growth has evolved significantly in modern economics, transitioning from human capital theory to endogenous growth theories emphasizing the pivotal roles of skills, knowledge, and innovation. While higher education (HE) has long been recognized as a primary incubator for advanced knowledge and skilled talents, catalyzing regional innovation and technology diffusion, existing research has predominantly focused on macroscopic levels, overlooking peripheral regions at the county level.

Since 1999, China’s expansion policy of HE enrollment has led to the establishment of rural university campuses, prompting inquiries into their impact on economic growth and regional disparities across China’s regions. Utilizing panel data spanning 20 years from nine cities across China (collecting “a total of 58 counties and 156 rural university campuses from the nine cities, representing 53.8% of all rural university campuses” in China, see lines 181-185), this study examines the impact of rural university campuses on county economic growth.

Literature explored by the authors suggests a causal relationship between the decentralization of HE and regional economic development (see lines 110-111), with the impact of rural university campuses strengthening over time. Studies by Cantoni and Yuchtman, Boucher et al., Benneworth and Charles, and Kantor and Whalley support this assertion, highlighting the crucial role of universities in expanding economic activities and enhancing local business productivity.

The study’s findings reveal that rural university campuses have a positive effect on county economic growth, especially in the secondary and tertiary sectors, as hypothesized by the authors (H1, lines 134-138). These campuses play a significant role in fostering knowledge-intensive business services and facilitating social activities within the region. However, their impact on agriculture appears to be limited. Moreover, the study underscores the strengthening influence of these campuses over time, coinciding with the development of human capital and the enhanced employability of graduates (lines 139-140). This temporal effect aligns with the authors’ hypothesis (H2, lines 155-156), emphasizing the gradual strengthening of county economic development over time.

The authors conduct a longitudinal analysis to compare economic performance at different time points following the establishment of rural university campuses. They use regression analysis with fixed effects for each year post-establishment, incorporating dummy variables to capture time effects. In Table 4 (line 295), regression coefficients for various economic indicators are estimated for each time point (Year 1, Year 2, Year 3), including GDP, value added across different industries, fixed investment, and budget revenue. Year fixed effects control for time-specific factors, while county fixed effects address time-invariant county characteristics. This approach ensures that observed differences in outcomes between counties with and without campuses are not solely attributed to unobserved factors.

The approach of expansion of time-varying DID Model expands upon the initial time-varying Difference-in-Differences (DID) model used in the analysis. DID models are commonly used in econometrics to estimate the causal effect of a treatment or intervention by comparing changes in outcomes over time between a treatment group (affected by the intervention) and a control group (not affected). In this case, the binary variables representing the adoption of rural university campuses serve as the treatment indicator, allowing for the estimation of their impact over time.

The regression results provide insights into how the impact of rural university campuses on economic performance evolves over time. By examining the coefficients associated with rural university campuses at different time points, the authors can assess whether their effects on economic indicators strengthen, weaken, or remain stable over time.

Regional disparities are evident, with rural university campuses in the eastern region demonstrating a more pronounced impact on economic growth compared to those in the central and western regions. While hypothesis H3 touches upon this notion (lines 170-171), the authors could provide further clarity on whether this "regional heterogeneity" translates to improved employability prospects for students attending rural university campuses. Although the study suggests a significant role for rural university campuses in county-level economic development and provides a theoretical framework and robust measurement tools to tackle endogeneity issues, the discussion on the relationship between the campuses' significance and the employability of their graduates (as mentioned in section 5, lines 350-352) relies heavily on secondary sources and would benefit from additional elaboration.

The authors utilize robust econometric techniques such as the ‘Synthetic Control Method,’ which allows them to construct counterfactual scenarios for rural areas that did not have the intervention of rural university campuses. This approach helps us compare the economic growth trajectories of treated and untreated rural areas, thereby controlling for potential confounding factors and providing more accurate estimates of the impact of rural university campuses on economic development (see lines 239-253). The authors finally propose policy recommendations to enhance the positive impact of rural university campuses on county-level economic development. These recommendations include coherent supportive policies jointly introduced by national, regional governments, and rural university campuses, the establishment of adequate supporting institutions around rural campuses, and the introduction of rural university campuses based on regional disparities in economic development.

However, the study acknowledges limitations, including the limited representativeness of the sample covering rural university campuses in nine major cities in China and the failure to fully account for the effects of different types of county-level campuses. Future research is encouraged to explore these aspects more extensively with higher-quality data.

Advised amendments:

1.      While your study primarily focuses on the establishment and development of rural university campuses as the independent variable, Benneworth et al. (2022) emphasize the role of these campuses in “mobilizing local learning communities,” which in turn contributes to economic growth, especially in secondary and tertiary sectors.

Could you please provide insights into how the mobilization of local learning communities was considered or accounted for in your research methodology? Additionally, do you think this broader perspective on the impact of rural university campuses could influence the interpretation of your findings regarding their effect on regional economic growth?

2.      In Section 2, it is crucial to provide a comprehensive overview of the sectors developed in rural regions of China, particularly within the nine counties considered in this study. Understanding the economic landscape of these regions is essential for contextualizing the analysis and interpreting the findings accurately. In rural China, economic activities often revolve around agriculture, manufacturing, services, and emerging industries. Agriculture remains a fundamental sector, with a focus on crop cultivation, livestock farming, and agribusiness. In addition to traditional agriculture, rural areas have witnessed a growing presence of agro-processing industries, contributing to value addition and rural employment. Manufacturing activities in rural China encompass a range of industries, including light manufacturing, textiles, food processing, and small-scale production units. Services play an increasingly significant role, driven by the expansion of rural markets and the demand for various services such as retail, healthcare, education, and tourism. Moreover, emerging sectors such as renewable energy, e-commerce, and technology-based enterprises are gaining traction, presenting new opportunities for rural economic development. By delineating the sectors developed in rural regions of China, this study can elucidate the specific pathways through which rural university campuses influence economic growth and structural transformation, providing valuable insights for policymakers and stakeholders invested in promoting inclusive and sustainable development across different regions of China.

3.      The study acknowledges that different types of county-level campuses may have varying effects (see lines 424-428). To address this, the authors should have included an analysis that accounts for factors such as intra-city relocation, construction in different locations, and synchronous education between urban and rural areas to provide a more nuanced understanding of the impact of rural university campuses. Since accounting for such diverse factors could indeed be challenging, instead of attempting to analyze all these factors comprehensively, the authors could consider conducting a preliminary exploratory analysis to identify the most influential factors among those mentioned. Based on this analysis, they could prioritize the inclusion of the most significant factors in their model, while acknowledging the limitations and complexities of capturing the full range of campus types and their effects. This approach would allow for a more focused and manageable analysis while still addressing the heterogeneity of campus types to some extent.

4.      Terms at lines 203-210 should be clarified for better understanding, example:

o    Quantity of operating Rural university campuses in county n in year t: This refers to the number of rural university campuses that are actively operating within a specific county (denoted as n) during a particular year (denoted as t). These campuses are the ones that are open and functioning, providing educational services and engaging in various activities related to their mission.

o    Cumulative operational years of Rural university campuses in county n: This represents the total duration for which rural university campuses have been operational within a specific county (n) up to the current year (t). It is the sum of the years since each campus was established and began its operations in the county. For example, if a campus was established five years ago and has been operating continuously since then, its cumulative operational years would be five.

5.      In your paper, you presented regression analysis results in Table 2 line 263, providing descriptive statistics for various variables. Could you please clarify the software or tools you utilized to conduct the regression analysis? Did you employ specialized statistical software like Stata, R, or Python libraries, or did you use spreadsheet software like Excel for the calculations?

Understanding the software used for the analysis would provide valuable insights into the methodological approach and facilitate transparency and reproducibility of your findings.

6.      In Table 5 line 313, the interaction term 'Rural university campuses (Year 1/4/6) x Eastern area’ is utilized to examine the differential impact of rural university campuses between the eastern region and other regions (central and western) of China. Could you please clarify what the ‘x’ represents in this context? Is it denoting a categorical interaction between the presence of rural university campuses and the eastern region, or does it stand for another factor? Additionally, could you explain how the coefficient associated with this interaction term quantifies the difference in impact between the eastern region and the reference regions?

7.      In your study, you propose that rural university campuses serve as primary drivers of county-level economic development (see lines 370-371). Could you please provide some examples or insights into what specific factors or mechanisms contribute to this role as primary drivers of economic development in the areas under study?

8.      Your research methodology appears to incorporate various statistical techniques such as causal relationship analysis, Difference-in-Differences (DID) modeling, Moretti’s multiplier, and the Synthetic Control Method. However, the transition between these methods is not clearly delineated, which may obscure the overall framework of your study. Could you please consider reformulating your methods section to provide a clearer and more orderly explanation of how each statistical technique contributes to your analysis? This would help readers better understand the logical flow of your research methodology.

9.      While the policy recommendations provided are insightful, the authors could further enhance them by offering specific actionable steps that policymakers can take to implement these recommendations effectively. This could include detailed strategies for coordinating supportive policies, establishing supporting institutions, and targeting the placement of rural university campuses based on regional disparities.

By addressing these points, the authors can strengthen the robustness and applicability of their study, providing valuable insights for policymakers and researchers in the field of rural development and higher education.

References for the review:

Benneworth, P., Maxwell, K., Charles, D. (2022). Measuring the effects of the social rural university campus. Research Evaluation, rvac027, pp. 1-10. https://doi.org/10.1093/reseval/rvac027

Valero, A., Van Reenen, J. (2019). The economic impact of universities: Evidence from across the globe, Economics of Education Review, Volume 68, pp. 53-67, ISSN 0272-7757. https://doi.org/10.1016/j.econedurev.2018.09.001

Author Response

Reviewer 2

  1. While your study primarily focuses on the establishment and development of rural university campuses as the independent variable, Benneworth et al. (2022) emphasize the role of these campuses in “mobilizing local learning communities,” which in turn contributes to economic growth, especially in secondary and tertiary sectors.

Could you please provide insights into how the mobilization of local learning communities was considered or accounted for in your research methodology? Additionally, do you think this broader perspective on the impact of rural university campuses could influence the interpretation of your findings regarding their effect on regional economic growth?

Response:

We are deeply grateful for the reviewer’s detailed and insightful comments on our manuscript. The focus of this study was to investigate the economic impact of decentralized higher education institutions on local economic development. We did not dive into the moderating or mediating role of “mobilization of local learning communities”. Nevertheless, the reviewer has raised a very interesting and useful perspective into the relationship between university campuses and local economy.   

  1. In Section 2, it is crucial to provide a comprehensive overview of the sectors developed in rural regions of China, particularly within the nine counties considered in this study. Understanding the economic landscape of these regions is essential for contextualizing the analysis and interpreting the findings accurately. In rural China, economic activities often revolve around agriculture, manufacturing, services, and emerging industries. Agriculture remains a fundamental sector, with a focus on crop cultivation, livestock farming, and agribusiness. In addition to traditional agriculture, rural areas have witnessed a growing presence of agro-processing industries, contributing to value addition and rural employment. Manufacturing activities in rural China encompass a range of industries, including light manufacturing, textiles, food processing, and small-scale production units. Services play an increasingly significant role, driven by the expansion of rural markets and the demand for various services such as retail, healthcare, education, and tourism. Moreover, emerging sectors such as renewable energy, e-commerce, and technology-based enterprises are gaining traction, presenting new opportunities for rural economic development. By delineating the sectors developed in rural regions of China, this study can elucidate the specific pathways through which rural university campuses influence economic growth and structural transformation, providing valuable insights for policymakers and stakeholders invested in promoting inclusive and sustainable development across different regions of China.

Response:

We appreciate the reviewer's feedback on the insufficient content of the literature review. In Section 2, we have now included an introduction to the development sectors of rural China on pages 2-3, as follows:

"2.1. Overview Of the Economic Sectors in Rural China

In rural China, economic activities typically revolve around agriculture, manufacturing, services, and emerging industries. Agriculture remains a fundamental sector, focusing on crop cultivation, animal husbandry, and agricultural enterprises. After 20 years of economic reforms, significant growth has been achieved in rural China. Despite concerns in the early 1990s about the slowdown in total factor productivity growth after a rapid increase in the early 1980s, by the late 1980s and 1990s, per capita food supply in China had reached levels similar to those of developed countries, making China one of the world's fastest-growing food exporting countries [9]. Economic activities in rural China are not limited to the productivity of the agricultural sector alone; in addition to traditional agriculture, the rural areas witness an increasingly diverse range of agricultural processing industries and emerging commodity markets, contributing to value addition and rural employment. Manufacturing activities in rural China encompass a range of industries including light industry, textiles, food processing, and small-scale production units. Despite attempts by the government to control the market over the past decade, agricultural markets have not only emerged but have also thrived, resembling markets in other countries more and more [11].

In addition, the transition of population from rural to urban areas and from agriculture to industry is also central to its economic development [12]. Some scholars consider the transformation of population structure in China in the 1980s to be lagging behind income levels [13]. However, in recent years, the transformation of the non-agricultural sector has been faster than any other sector. In the mid-1990s, barriers between rural and urban areas broke down, initiating an unprecedented and perhaps irreversible flow of labor to cities. Despite favorable macroeconomic conditions in the late 1990s, the surge in non-agricultural employment not only continued after 1995 but also accelerated. Today, over 200 million people work outside farms, with over half of them working in cities. Nearly 85% of rural households have at least one member working in the non-agricultural sector [14].

Moreover, emerging industries such as renewable energy, e-commerce, and technology-based enterprises are gaining increasing attention, providing new opportunities for rural economic development [15]. In recent years, the rapid development of e-commerce in China's agricultural products has largely played a role in linking and matching the digital economy, accelerating the integration of agriculture and industrial services [16]. Digital finance in rural China not only enhances the production function of agriculture but also adds a leisure function, demonstrating good performance in the integration of agriculture and services [17]. However, at present, within the Chinese government, there is a lack of colleagues who understand information technology, possess management skills, and have rich rural work experience, and there is a shortage of suppliers in China's digital agricultural construction. The development level of information infrastructure in different regions is unbalanced, the construction of data sharing systems lags behind, severely hindering the integrated development of rural industries, and limiting the driving force of the digital economy for urban-rural integration [18-19]." (see on pages 2-3)

  1. The study acknowledges that different types of county-level campuses may have varying effects (see lines 424-428). To address this, the authors should have included an analysis that accounts for factors such as intra-city relocation, construction in different locations, and synchronous education between urban and rural areas to provide a more nuanced understanding of the impact of rural university campuses. Since accounting for such diverse factors could indeed be challenging, instead of attempting to analyze all these factors comprehensively, the authors could consider conducting a preliminary exploratory analysis to identify the most influential factors among those mentioned. Based on this analysis, they could prioritize the inclusion of the most significant factors in their model, while acknowledging the limitations and complexities of capturing the full range of campus types and their effects. This approach would allow for a more focused and manageable analysis while still addressing the heterogeneity of campus types to some extent.

Response:

We deeply appreciate the reviewer's insightful suggestion. While it might be tempting to incorporate factors like intra-city relocation, construction in various locations, and synchronous education between urban and rural areas, such aspects are beyond the scope of our current study. Our research, spanning from 2001 to 2020, is specifically aimed at investigating the longitudinal relationship between rural university campuses and county-level economic growth. While factors such as intra-city relocation and construction in diverse locations pertain to a horizontal perspective, which we did not cover, our study primarily emphasizes the longitudinal aspect. Although considering both longitudinal and horizontal perspectives would be ideal, such a comprehensive workload typically exceeds the capacity of a single study.

  1. Terms at lines 203-210 should be clarified for better understanding, example:

o Quantity of operating Rural university campuses in county n in year t: This refers to the

number of rural university campuses that are actively operating within a specific county (denoted as n) during a particular year (denoted as t). These campuses are the ones that are open and functioning, providing educational services and engaging in various activities related to their mission.

o Cumulative operational years of Rural university campuses in county n: This represents the total duration for which rural university campuses have been operational within a specific county (n) up to the current year (t). It is the sum of the years since each campus was established and began its operations in the county. For example, if a campus was established five years ago and has been operating continuously since then, its cumulative operational years would be five.

Response:

We are grateful to the reviewer for highlighting the issue of unclear terminology. In addition to revising the two examples provided by the reviewer, upon careful scrutiny of the entire manuscript, we realized that there was a lack of standardized interpretation for "Growthnt." Therefore, we have made the following modifications:

" Growthnt refers to the comprehensive economic growth and the influence of various sectoral economic indicators within a specific county (denoted as 'n') in a particular year (denoted as 't'). This includes factors such as the county's GDP, value added in primary, secondary, and tertiary industries, completed fixed asset investment, general public budget revenue, among others." (see page 6)

“(1) Quantity of operating rural university campuses in county n in year t: This refers to the number of rural university campuses that are actively operating within a specific county (denoted as n) during a particular year (denoted as t). These campuses are the ones that are open and functioning, providing educational services and engaging in various activities related to their mission. (2) Cumulative operational years of rural university campuses in county n: This represents the total duration for which rural university campuses have been operational within a specific county (n) up to the current year (t). It is the sum of the years since each campus was established and began its operations in the county. For example, if a campus was established five years ago and has been operating continuously since then, its cumulative operational years would be five.” (see page 6)

  1. In your paper, you presented regression analysis results in Table 2 line 263, providing descriptive statistics for various variables. Could you please clarify the software or tools you utilized to conduct the regression analysis? Did you employ specialized statistical software like Stata, R, or Python libraries, or did you use spreadsheet software like Excel for the calculations?

Understanding the software used for the analysis would provide valuable insights into the methodological approach and facilitate transparency and reproducibility of your findings.

Response: 

We are thankful for the reviewer’s suggestion. The follow information has now been added to Section 3.2 on page 5:

“This study primarily employed the Stata software for data analysis.”

In Table 5 line 313, the interaction term 'Rural university campuses (Year 1/4/6) x Eastern area’ is utilized to examine the differential impact of rural university campuses between the eastern region and other regions (central and western) of China. Could you please clarify what the ‘x’ represents in this context? Is it denoting a categorical interaction between the presence of rural university campuses and the eastern region, or does it stand for another factor? Additionally, could you explain how the coefficient associated with this interaction term quantifies the difference in impact between the eastern region and the reference regions?

Response:

We are thankful for the reviewer’s suggestion. The “x” represents the interaction between two variables. The following information has now been added to Section 4.4 on page 10:

We introduced the interaction term of "rural university campuses (Year 1/4/6) x eastern area" to examine the differential effects of rural university campuses between the eastern and central-western regions of China. Furthermore, by incorporating a time axis, this study examined the long-term variations of these effects. The sign and magnitude of the interaction term could help us understand the interaction between these variables. If the interaction term of two independent variables is positive, then their combined effect will be greater than the sum of their individual effects.” (see page 10)

  1. In your study, you propose that rural university campuses serve as primary drivers of county-leveleconomic development (see lines 370-371). Could you please provide some examples or insights into what specific factors or mechanisms contribute to this role as primary drivers of economic development in the areas under study?

Response:

We appreciate the reviewer’s suggestion. We have added the following information to page 12:

While the overall level of local attachment and entrepreneurial willingness among university graduates remains relatively low [49], county-level universities play a significant role in augmenting the local labor force directly and indirectly. Studies indicate that the prosperity of talent serves as a critical driver for the comprehensive development of rural industries [50]. Human capital emerges as a potent force in facilitating the integration of rural industries [51]. Mechanism studies reveal that in areas with low levels of rural human capital and correspondingly lower cultural proficiency among rural laborers, there exists a challenge of limited internet awareness and digital technology application capabilities. This impedes rural laborers from accessing vital information on agricultural industry development through digital platforms and complicates the seamless integration of agricultural digital technology with other sectors. Conversely, regions with higher levels of rural human capital and enhanced cultural proficiency among rural laborers are better equipped to leverage agricultural digital technology and gather agricultural resource information, thus fostering the holistic development of rural industries [52].” (see page 12)

  1. Your research methodology appears to incorporate various statistical techniques such as causal relationship analysis, Difference-in-Differences (DID) modeling, Moretti’s multiplier, and the Synthetic Control Method. However, the transition between these methods is not clearly delineated, which may obscure the overall framework of your study. Could you please consider reformulating your methods section to provide a clearer and more orderly explanation of how each statistical technique contributes to your analysis? This would help readers better understand the logical flow of your research methodology.

Response:

We appreciate the reviewer’s insightful feedback. As suggested, the following information has now been added to Section 3.2 on pages 5 and 6:

We initially employed a time-varying Difference-in-Differences (DID) model due to variations in the timing of the establishment of university campuses in counties across different cities in China's central and western regions, as well as differences in the number of county-level campuses in each city. Traditional double-difference methods generally apply to situations where the sample status remains unchanged at the same time point. Additionally, we used event-study analysis to observe the differences in effects before and after the implementation of rural university campuses in two sample groups. This analysis also captured dynamic changes over time and validated Hypothesis 2. Finally, synthetic control methods were employed to determine weights through data-driven approaches, reducing subjective selection errors and avoiding endogeneity issues in policies.” (see pages 5-6)

  1. While the policy recommendations provided are insightful, the authors could further enhance them by offering specific actionable steps that policymakers can take to implement these recommendations effectively. This could include detailed strategies for coordinating supportive policies, establishing supporting institutions, and targeting the placement of rural university campuses based on regional disparities.

Response:

We appreciate the reviewer's reminder to enhance the recommendation section. We have now included the following sentences addressing coordinated support policies, establishment of support institutions, and rural university campus differentiation based on regional disparities:

Policies should consider how to encourage collaborative research formats to establish a common understanding among academia, rural small and medium-sized enterprises, civil society, and government. Specialized systems related to the distinctive development of rural university campuses should be promptly introduced to clarify the development goals and positioning of rural university campuses, as well as delineate relevant institutional frameworks for their distinctive development, thereby providing theoretical frameworks and practical bases for promoting characteristic development in rural areas and rural university campuses.” (see page 13)

For university administrators and policymakers, it is essential to recognize that besides personalized benefits for each student, community interests constitute a significant aspect of these campuses. While every student benefits from their higher education experience, the model of rural university campuses attempts to determine how communities can also benefit alongside students, even if students leave the area upon graduation. Student knowledge exchange becomes an additional avenue to create broader community value, fostering learning among community members, building social capital and community resilience, and ensuring sustained benefits.” (see page 14)

We should increase investment in rural university campuses; actively promote the development advantages of rural university campuses; integrate rural university campuses into the public service system of large, medium, and small cities, appropriately expanding the overall management authority of rural university campuses; increase funding for rural university campuses and implement strong-weak paired assistance policies. To address discrepancies in educational concepts and operating methods, developed provinces can assist rural university campuses in other provinces, encouraging provinces (regions, municipalities) to organize local universities to conduct custodial assistance work to achieve win-win cooperation.”(see page 14)

The government needs to consider the overall situation, refine its top-level design, and formulate sound policies to guide the integrated development of rural university campuses and local economic industries based on regional disparities and particularities. It should establish a community of rural university campuses, set up demonstration areas and schools, actively encourage mutual learning and communication among regions, rationally plan the coordinated construction of rural university campuses between regions, encourage the sharing of similar regional information and resources, and facilitate win-win cooperation between complementary regions, thus ensuring the smooth flow of knowledge capital and technology among regions.” (see page 14)

Reviewer 3 Report

Comments and Suggestions for Authors

The analysis proposed by authors aims at investigating, with reference to the Chinese context, the connection between the presence of rural university campuses and the county-level GDP and industrial composition. The analysis covered a period of 20 years. The manuscript is, in general, well-structured and the methodology used is thoroughly presented. According to the same methodology, the whole discourse is conducted in an orderly and logical manner. As a consequence, results and discussion are consistently supported. Policy implications highlight interesting issues and complete the analysis conducted.

Revisions are as follows:

Title – authors should not use the term relationships rather than “impact” or “influence” of rural university campuses on rural economic development. The word evidence should not be in capital letters as it is placed after colon.

Line 7 – the phrase “These authors contributed equally to this work” should be deleted and placed in Line 430 Author Contributions.

Line 13 “which started” refers to universities although it is right after the word “source”. Please rewrite accordingly.

Part 2 is meant to be Literature review although it embeds, starting from part 2.2, research hypotheses. Please rearrange accordingly.

Author Response

Reviewer 3

Comments and Suggestions for Authors

The analysis proposed by authors aims at investigating, with reference to the Chinese context, the connection between the presence of rural university campuses and the county-level GDP and industrial composition. The analysis covered a period of 20 years. The manuscript is, in general, well-structured and the methodology used is thoroughly presented. According to the same methodology, the whole discourse is conducted in an orderly and logical manner. As a consequence, results and discussion are consistently supported. Policy implications highlight interesting issues and complete the analysis conducted.

Revisions are as follows:

Title – authors should not use the term relationships rather than “impact” or “influence” of rural university campuses on rural economic development. The word evidence should not be in capital letters as it is placed after colon.

Response:

We are grateful for the reviewer’s suggestion. Accordingly, the title has been revised as follows:

“Investigating the influence of rural university campuses on rural economic development: evidence from Chinese counties between 2001 and 2020”

Line 7 – the phrase “These authors contributed equally to this work” should be deleted and placed in Line 430 Author Contributions.

Response:

We are thankful for the reviewer’s kind suggestion. This phrase has now been deleted from line 7 and relocated it to the section of Author Contributions.

Line 13 “which started” refers to universities although it is right after the word “source”. Please rewrite accordingly.

Response:

We appreciate the reviewer’s kind suggestion. The sentence has now been rewritten, as follows:

“Utilizing the data of the Chinese universities that started to establish their campuses in counties since the year of 1999, this study investigates the influence of rural university campuses on county-level GDP and industrial composition spanning from 2001 to 2020.” (see page 1)

Part 2 is meant to be Literature review although it embeds, starting from part 2.2, research hypotheses. Please rearrange accordingly.

Response:

We are thankful for the reviewer’s suggestion. To make the heading of Part 2 clearer, we have now modified it from “Literature Review” to “Literature Review and Hypotheses Development”. (see page 2)

Round 2

Reviewer 2 Report

Comments and Suggestions for Authors

Subject: Revised version of paper ID: Sustainability_2956678

Thank you for your thoughtful response to my comments on your manuscript. I appreciate the depth and clarity you've added to the discussion, particularly regarding the economic landscape of rural China and the impact of rural university campuses on regional economic growth. Your revised manuscript provides a clearer picture of the diverse economic activities shaping these regions, crucial for understanding how these institutions influence economic growth and structural transformation.

The additional information about the economic indicators used in your study, including the standardized interpretation of "Growthnt," enhances understanding of the county-level analysis and the longitudinal impact of rural university campuses. The methodological rigor you've shown, especially with the inclusion of Stata for data analysis, adds credibility and aids in the reproducibility of your findings.

I found the detailed responses regarding the methodological choices, such as the use of the Difference-in-Differences (DID) model and the Synthetic Control Method, particularly insightful. These methods, along with the event-study analysis, appear well-chosen to capture the dynamic impacts of rural university campuses over time and reduce biases related to subjective selection and endogeneity.

Your manuscript now effectively addresses the direct economic impacts of decentralized higher education institutions without delving into mediation or moderation effects, which you noted fall outside the scope of this study. However, your engagement with the concept of mobilizing local learning communities and its potential impact on economic growth is commendable and enriches the narrative.

The discussion on the challenges and opportunities in integrating digital technology into rural industries, alongside the role of human capital, provides a nuanced understanding of the barriers to economic growth in rural areas. This context is vital for comprehending the strategic importance of rural universities in fostering regional development.

Suggestions for further improvement include:

·        Implementation Timelines: Detailing when and how the suggested actions might be implemented could enhance the practicality of your recommendations.

·        Metrics for Success: Articulating specific metrics or indicators of success could facilitate better monitoring and adjustment of the strategies you propose.

·        Stakeholder Roles and Responsibilities: A clearer delineation of roles for various stakeholders would aid in better coordination and implementation of policies.

·        Case Studies or Examples: Including practical examples or case studies of successful strategies could serve as a blueprint and inspire confidence in your approaches.

·        Challenges and Risks: Addressing potential challenges and proposing measures to mitigate them would reflect a deeper understanding of the complexities involved.

Overall, your commitment to enhancing the clarity and depth of your research is evident and commendable. The manuscript significantly benefits from the comprehensive and actionable strategies you have outlined, which not only address my initial concerns but also enhance the scholarly contribution of your work. I look forward to the continued evolution of your research and its impact on the field.

Best regards,

Author Response

We are deeply grateful for the reviewer's detailed and insightful feedback during this round of review. As suggested, the following paragraph has now been added to "Section 6. Policy Implications":
"In sum, the proposed policies stemming from this study's findings are poised for prompt implementation, with initial planning and coordination among national, regional governments, and rural university campuses expected to span 6-12 months. Success could be gauged through a diverse array of metrics, including economic growth indicators, graduate return rates, collaborative efforts, and enhancements in talent retention and community engagement. Stakeholders, ranging from governments to rural universities, academia, small and medium-sized enterprises, and civil society, each have distinct yet interrelated roles in policy formulation, implementation, and collaboration, necessitating clear delineation of responsibilities. Drawing upon successful case studies from other regions, such as rural development programs in Scandinavia or initiatives in rural revitalization in the United States, could provide valuable insights and inspirations for effective strategies, challenges encountered, and lessons learned. However, addressing coordination hurdles between stakeholders, ensuring equitable resource distribution, and adapting policies to diverse regional contexts are key challenges that require mitigation strategies like regular evaluations, capacity building, and targeted interventions. Additionally, fostering awareness, stakeholder engagement, and flexibility in policy design and implementation will be vital in navigating potential obstacles and ensuring the success and sustainability of the proposed policies." (see page 15)